# Untargeted Lipidomics Reveal Quality Changes in High-Moisture Japonica Brown Rice at Different Storage Temperatures

**DOI:** 10.3390/foods12234218

**Published:** 2023-11-22

**Authors:** Lingyu Qu, Yan Zhao, Xiangdong Xu, Yanfei Li, Haoxin Lv

**Affiliations:** 1School of Food and Strategic Reserves, Henan University of Technology, Zhengzhou 450001, China; 2021920031@stu.haut.edu.cn (L.Q.); liyanfei@haut.edu.cn (Y.L.); lvhaoxin0129@126.com (H.L.); 2Yihai Kerry (Wuhan) Oils & Grains Industries Co., Ltd., Wuhan 430040, China; xxd2755@126.com

**Keywords:** high-moisture japonica brown rice, storage temperature, color, fatty acid value, volatile compound, metabolism pathway

## Abstract

Low temperatures are an effective way of delaying grain rancidity and deterioration. However, little is known about the difference in quality changes in high-moisture japonica brown rice at different storage temperatures. In this study, the storage quality changes in japonica brown rice with a 15.50% moisture content stored at 15 °C, 20 °C, and 25 °C were investigated. In addition, an untargeted lipidomics analysis coupled with gas chromatography and mass spectrometry (GC-MS) was applied to analyze the volatile compounds and metabolite changes in the high-moisture japonica brown rice. The results showed that storage at 15 °C could well maintain the color and aroma stability of the brown rice and delay the increase in fatty acid value (FAV). The lipidomics results showed that storage at 15 °C delayed glycerolipid and sphingolipid metabolism and reduced glycerophospholipid catabolism in the brown rice. The low-temperature environment regulated these three metabolic pathways to maintain higher contents of triglycerides (TG), phosphatidylserine (PS), abd phosphatidylethanolamine (PE), and lower contents of diglycerides (DG), OAcyl-(gamma-hydroxy) FA (OAHFA), ceramides (Cer), and glycosylceramides (Hex1Cer) in the high-moisture japonica brown rice, which maintained the storage stability of the brown rice. Our results proposed the cryoprotection mechanism of postharvest brown rice from the perspective of volatile compounds and metabolite changes, providing a foothold for the further exploration of low-temperature storage as a safe and efficient cryoprotectant in the grain storage field.

## 1. Introduction

Rice is a dietary staple for over half of the world’s population [1], with harvested rice often being processed into either white rice or brown rice for consumption. The two main types of rice processing are as follows: firstly, processing rice by removing the hull from the paddy and producing brown rice, and secondly, processing brown rice by removing the bran and then polishing it to obtain white rice [2]. Brown rice has become increasingly popular among consumers due to its rich content of dietary fiber, vitamins, gamma-aminobutyric acid (GABA), gluten, and fats that are beneficial to one’s health [3].

The moisture content of brown rice during storage affects its quality changes. However, it has been shown that high-moisture brown rice has a better freshness and taste, such that cooked rice grains have the best texture when stored with a 15.70% moisture content [4]. In addition, the storage environment is also one of the factors affecting the quality of brown rice storage. Brown rice under humidity and temperature conditions conducive to mold growth can lead to the rapid colonization of spoilage molds, mycotoxin contamination, and quality loss [5]. Therefore, a series of brown rice treatment and storage methods have emerged. For instance, low-temperature storage delays the increase in the fatty acid values of red brown rice [6]. Infrared drying has shown positive effects on the color, microstructure, cooking, texture gelatinization, and pasting characteristics of stored brown rice [7]. High-pressure treatment (especially at 200 MPa for 10 min) was found to be efficient to reduce rancidity in brown rice during storage, which could lead to large economic benefits [8]. However, changes in lipids and color of high-moisture japonica brown rice storage at different temperatures are not clear.

Gas chromatography-mass spectrometry (GC-MS) is an important technical method widely used for the identification and quantification of volatile compounds in sample matrix mixtures [9]. Due to the loss of shell protection, brown rice is more susceptible to the oxidative decomposition of its lipids to produce compounds such as glutaraldehyde and undesirable odors compared to rice, thus affecting the taste quality of brown rice. Liu et al. [10] investigated with the GC-MS technique and found that the composition of volatile compounds in brown rice is complex, including hydrocarbons, alcohols, acids, esters, aldehydes, ketones, phenols, and others. Zhou et al. [11] found that a few trace flavor components were only present in some rice samples using a GC-MS technique analysis, but they had high-correlation coefficients with a variety of flavor compounds, which provided a special aroma to the rice. Importantly, storage temperature and time significant influenced the content of volatiles in brown rice [12]. Therefore, it is necessary to further investigate the changes in volatile flavor compounds and their correlations in high-moisture Japonica brown rice at different storage temperatures.

Lipidomics is a critical approach for studying lipid metabolism and related metabolic pathways [13]. Until now, lipidomics have been widely applied to analyze lipid component profile analyses for food processing research, food preservation research, food nutrition, and health research [14], as well as food storage [15]. For example, Feng et al. [16] revealed the trends of lipid subclasses, FAV, total fatty acids, and aliphatic aldehydes in pork with freezing time using untargeted lipidomics and metabolomics techniques. Concepcion et al. [17] applied untargeted lipidomics technology to rice research and revealed the role of specific lipids in rice texture. Meng et al. [18] analyzed the changes in egg yolk lipids during storage using untargeted lipidomics. Similarly, lipidomics could be used to analyze changes in the lipid metabolism in brown rice during storage [19]. However, an untargeted lipidomic analysis has not been applied to analyze the lipidomic profile of high-moisture japonica brown rice at different storage temperatures.

The main objective in this study is to elucidate the impact of storage temperature on the quality deterioration of high-moisture japonica brown rice through an untargeted lipidomics approach. High-moisture japonica brown rice was stored at 15 °C, 20 °C, and 25 °C for 90 days, and the color, fatty acid value, volatile compounds, and lipid metabolites of the high-moisture japonica brown rice were investigated during storage at different temperatures. Significant differences in the lipid metabolites of the high-moisture japonica brown rice stored for 90 days were screened out and a metabolic pathway analysis was performed based on these lipid metabolites. The aim of this study is to provide a scientific theoretical basis for the low-temperature storage of high-moisture japonica brown rice.

## 2. Materials and Methods

### 2.1. High-Moisture Japonica Brown Rice Sample Preparation

Fresh japonica brown rice (Sui-Japonica 18) was collected from Yihai Kerry Oils & Grains Industries Co., Ltd. (Jiamusi City, Heilongjiang Province, China) in August 2022. Aseptic water was sprayed on the surface of the japonica brown rice, and the japonica brown rice was turned so that the aseptic water evenly adhered to the surface of the japonica brown rice. It was equilibrated at 25 °C overnight. The moisture content was measured the next day and the above steps were repeated until the moisture content of the brown rice was adjusted to 15.5 ± 0.2%, which will be used as high-moisture japonica brown rice for storage. High-moisture Japonica brown rice samples were packed in sealed polyethylene bags and stored in a constant temperature and humidity incubator (Ningbo Southeast Instrument Co., Ltd., Ningbo, China) for 90 days with temperatures of 15, 20, and 25 °C, with a relative humidity of 65% (Appendix A). Each treatment was repeated in three replicates. Samples were taken every 15 days and three parallel samples were taken at each sampling. All the samples were stored at −80 °C for further analysis.

### 2.2. Color and Fatty Acid Values Measurement

Colorimetric parameters were determined using the method of Wu et al. [20]. The skin color parameters of the bulk high-moisture japonica brown rice were detected using an NR200 colorimeter (Shenzhen 3NH Technology Co., Ltd., Shenzhen, China), and untreated samples were used as controls. In addition, the white reference was used for color measurements. The colorimetric parameters L* (lightness), a* (red-green), and b* (yellow-blue) were collected.

The fatty acid values (FAV) were measured according to the method of Qu et al. [21], with slight modification. The brown rice samples were ground into powder using a grinder (BLH-560KL, Bethlehem Apparatus Co., Ltd., Zhejiang, China). Fatty acids were extracted from the brown rice flour using benzene and the extract was titrated with KOH (0.01 mol/L) using a 0.04% phenolphthalein ethanol solution as an indicator. The result was represented as the number of milligrams of KOH required to neutralize the free fatty acids in 100 g of brown rice flour.

### 2.3. HS-SPME/GC-MS Analysis of Volatile Compounds

#### 2.3.1. Extraction of Volatile Compounds

The method of Li et al. [22] was used to extract the volatile compounds from the high-moisture japonica brown rice. Briefly, 15 g of high-moisture japonica brown rice was taken inti a 60 mL brown headspace bottle and the bottle was sealed with plastic wrap. Subsequently, it was placed in a constant temperature water bath at 70 °C for 1 h. The extraction head was then inserted into the headspace bottle for 70 min, after which, it was resolved in the GC-MS injection port (250 °C, non-shunt mode) for 5 min. The extraction head used in this experiment was an SPME fiber coated with divinylbenzene/carboxy/polydimethylsiloxane (DVB/CAR/PDMS, 50/30 μm).

#### 2.3.2. Setting of Gas Chromatography (GC) Conditions

Referring to the method of Li et al. [23] with slight modifications, a HP-5MS capillary column (30 m × 0.25 mm, 0.25 μm) was used. Column warming procedure: the initial temperature was 45 °C and maintained for 5 min, the temperature was increased to 250 °C at 5 °C/min, and maintained at 250 °C for 5 min; the whole heating procedure ran for 51 min. The carrier gas was high-purity helium at a flow rate of 1.0 mL/min (99.999%), and the sampling method was a non-split flow injection.

#### 2.3.3. Setting of Mass Spectrometry (MS) Conditions

The inlet temperature was 250 °C, the interface temperature was 280 °C, the ion source was an EI source and its temperature was 230 °C, the quaternary rod mass spectrometer and the quaternary rod temperature was 150 °C, and the electron energy source was 70 eV. The information was collected using full scanning mass spectrometry with a mass scanning range of 50–550 *m*/*z*, and the solvent delay time was 1 min.

#### 2.3.4. Identification and Relative Content Analysis of Volatile Compounds

The volatile compounds were preliminarily identified based on a comparison of the mass spectra obtained from the experiments with the standard mass spectral libraries of NIST08 and Wiley8, and characterized according to the degree of similarity; match fractions < 80% were used as cutoff values. The relative quantification of the components was based on the ion flow chromatographic peak area normalization method. The relative peak area percentage of each volatile compound was used as its concentration percentage.

### 2.4. Detection and Characterization of Lipid Metabolites

#### 2.4.1. Extraction of Lipid Metabolites

Accurately, 50 ± 5 mg of brown rice sample was weighed into a 2 mL centrifuge tube, and 6 mm diameter grinding bead, 280 µL of extraction solution (methanol:water = 2:5), and 400 µL of methyl tert-butyl ether were added. The mixed samples were ground for 6 min (−10 °C, 50 Hz) on a frozen tissue grinder and extracted using cryosonication for 30 min (5 °C, 40 KHz). Subsequently, the samples were allowed to stand at −20 °C for 30 min and centrifuged for 15 min (13,000× *g*, 4 °C). In total, 350 µL of supernatant was taken in an EP tube and blown dry with nitrogen. A total of 100 µL of extraction solution (isopropanol:acetonitrile = 1:1) was added and vortexed for 30 s. Finally, the samples were extracted using low-temperature ultrasonic extraction for 5 min (5 °C, 40 KHz) and centrifuged for 10 min (13,000× *g*, 4 °C), and the supernatant was pipetted into an injection vial with an internal cannula for analysis.

#### 2.4.2. UHPLC-LC-MS Analysis

The instrument platform for this LC-MS analysis was a Thermo Fisher Ultra High Performance Liquid Chromatography Tandem Fourier Transform Mass Spectrometry UHPLC-Q Exactive HF-X system. Chromatographic conditions: the chromatographic column was an Accucore C30 column (100 mm × 2.1 mm i.d., 2.6 µm; Thermo); the mobile phase A was a 50% acetonitrile aqueous solution (containing 0.1% formic acid and 10 mmol/L ammonium acetate) and the mobile phase B was acetonitrile/isopropanol/water (10/88/2) (containing 0.02% formic acid and 2 mmol/L ammonium acetate). The mobile phase B was acetonitrile/isopropanol/water (10/88/2) (containing 0.02% formic acid and 2 mmol/L ammonium acetate) with an injection volume of 5 μL. Mass spectrometry conditions: the samples were ionized using electrospray ionization, and the mass spectrometry signals were collected in positive and negative ion scanning modes, respectively. 

#### 2.4.3. Identification of Lipid Metabolites

The raw data were imported into the Lipidsearch 4.2.21 (Thermo Fisher, Waltham, MA, USA) for baseline filtering, peak identification, integration, retention time correction, and peak alignment, etc., and, finally, a data matrix containing the retention time, mass-to-charge ratio, and peak intensity information was obtained. Afterwards, the software was used to search the library for the identification of characteristic peaks, match the MS and MS/MS mass spectral information with the metabolic database, and the MS mass error was set to be less than 10 ppm, and the metabolites were identified based on the second-level mass spectral matching scores.

### 2.5. Statistical Analysis

The data were analyzed using a one-way analysis of variance (ANOVA) in SPSS 16.0 and *p* < 0.05 was considered to be a significant difference based on Duncan’s multiple range test. Each treatment was repeated in three replicates and all data were expressed as the means ± SDs.

Lipidomics analysis experiments using UHPLC-LC-MS were all performed four times in parallel. A Principal Component Analysis (PCA), orthogonal partial least squares-discriminate analysis (OPLS-DA), and cluster analysis were performed using Metaboanalyst 5.0 (https://www.metaboanalyst.ca/ (accessed on 1 July 2023)). The Kyoto Encyclopedia of Genes and Genomes (KEGG) database was used to analyze and match their corresponding lipid metabolic pathways.

## 3. Results and Discussion

### 3.1. Changes in Color of High-Moisture Japonica Brown Rice during Storage

Brown rice is more susceptible to color changes during storage due to the loss of protection of rice husk, which affects the appearance of brown rice as well as its acceptance [7]. As shown in Figure 1A–C, brown rice deterioration could be reflected more intuitively by color changes. The skin color of high-moisture japonica brown rice gradually turned dark and brown from yellow and quality deterioration appeared with a prolonged storage time, which could also be found with the changes in L*, a*, and b* values. At 90 days during storage, the high-moisture japonica brown rice stored at 15 °C showed significantly higher values of L*, a*, and b* than those at 20 °C and 25 °C, and the L*, a*, and b* in high-moisture japonica brown rice showed the significantly lowest levels at 25 °C. In addition, most of the fresh brown rice observed by the naked eye had a bright yellow color, while the aged brown rice had a dark yellow color (Figure 1D), revealing that low-temperature storage maintained a good color and luster, and delayed the quality deterioration of high-moisture japonica brown rice. Similarly, Kibar et al. [24] also found that the low-temperature storage of quinoa delayed its color darkening, in agreement with our results.

### 3.2. Changes in Fatty Acids Contents of High-Moisture Japonica Brown Rice during Storage

FAV is a measure of the free fatty acid content of fat, which can be used to indicate the degree of hydrolysis of fat during storage. Figure 1E showed that the FAV of high-moisture japonica brown rice at all three temperatures presented a rising tendency as the storage period increased. This could have been due to the high content of lipids in the embryo and seed coat of the brown rice, and the outer layer of brown rice was directly exposed to the external environment during storage [25]. Moreover, it is obvious that the FAV of the high-moisture japonica brown rice stored at 25 °C was higher than that of high-moisture japonica brown rice stored at lower temperatures (15 °C and 20 °C). Nevertheless, an excessive accumulation of fatty acids during storage would lead to rice rancidity and seriously degrade its edible quality [6].

Rice grains with an FAV of ≤25.0 mg/100 g are generally recognized as “suitable for storage”; Rice grains with 25.0 mg/100 g < FAV ≤ 35.0 mg/100 g are considered to be “mildly unfit for storage”; rice grains with FAV > 35.0 mg/100 g are recognized as being “heavily unfit for storage” [26]. In this study, the FAV of the high-moisture japonica brown rice already exceeded 25.0 mg/100 g when stored at 25 °C for 60 days. However, not only did the high-moisture japonica brown rice remain suitable for storage after 90 days at 15 °C, but the FAV of the high-moisture japonica brown rice at 15 °C was significantly lower than that of the high-moisture japonica brown rice at 20 °C and 25 °C (*p* < 0.05) (Figure 1E). It was found that a low temperature delayed the increase in FAV content in the rice stored at 15 °C and that a high storage temperature accelerated the aging of the rice grains [27]. Therefore, the high-moisture japonica brown rice stored at 15 °C could maintain its freshness better.

### 3.3. Analysis of Volatile Compounds in High-Moisture Japonica Brown Rice during Storage

Changes in the concentration of volatile compounds in high-moisture japonica brown rice during storage are shown in Table 1. A total of 38 volatile compounds were detected in all samples of high-moisture japonica brown rice, including alcohols (10), ketones (4), aldehydes (9), hydrocarbons (12), and others (3). Among these volatile compounds, esters were frequently characterized by a sweet, floral, and fruity aroma and aldehydes and alcohols contributed to green, grassy, fatty, fruity, and floral flavors [28]. Whereas most hydrocarbons exhibited no aroma, a few (such as dodecane and tridecane) produced an unpleasant gasoline odor. They may play a role in overall odor because of their high levels in rice [29]. Apparently, the high-moisture japonica brown rice had the greatest variety and percentage concentration of volatile compounds (except hydrocarbons) after 90 days of storage at 15 °C (Figure 1E and Table 1). This suggested that storage at 15 °C better preserved the volatile compounds that contributed to the aroma of high-moisture japonica brown rice, such as nonanal, 1-octanol, and (E)-2-nonenal, which imparted fruity and floral aromas to the high-moisture japonica brown rice [30]. In addition, 6,10,14-trimethyl-2-pentadecanone could give the high-moisture japonica brown rice a herbal and jasmine aroma [31].

The original volatile compounds in rice change with temperature and time during the storage period, such as decreasing or increasing in the content of certain original volatile compounds [32]. Although the concentrations of 1-octanol (fruit aroma), 4,8-dimethyl-1-nonanol, 5-ethyl-6-methyl-3E-hepten-2-one, 6,10-dimethyl-2-undecanone, nonanal (citrus aroma), and 2-nonenal (chicken aroma) decreased after 90 days of storage at 15 °C, they were still higher than the concentrations at 20 °C and 25 °C (Table 1). In addition, 2-pentylfuran was the most important alkylfuran identified in the brown rice with floral and nutty flavor characteristics [10]. The concentration of 2-pentylfuran in the high-moisture japonica brown rice increased by 1.08% after 90 days of storage at 15 °C, while it decreased at the other two temperatures. Interestingly, geranylacetone, which had a floral flavor in the high-moisture japonica brown rice, was detected only in the 15 °C rice, although its concentration decreased after 90 days of storage at 15 °C. It could be inferred that storage at 15 °C was more effective at maintaining the freshness and aroma stability of the high-moisture japonica brown rice.

### 3.4. Correlation Analysis of Color, FAV and Volatile Compounds

The correlation thermogram of color, FAV, and eleven volatile compounds of the high-moisture brown rice during storage is shown in Figure 2. L* exhibited a significant positive correlation with a* and undecanal, respectively. a* demonstrated a significant positive correlation with 2-hexyl-1-decanol and 2-pentylfuran, respectively. In addition, 1-octanol had a significant positive correlation with 1-nonanol, nonanal, 2,6,10-trimethyl-14-pentadecanone, undecanal, (E)-2-nonenal, and (E)-2-decenal, respectively. A significant positive correlation was also found between dodecane and tridecane, which had a gasoline odor.

Noticeably, FAV was negatively correlated with L*, a*, and b*, as well as 1-octanol, 1-nonanol, 2-hexyl-1-decanol, 2,6,10-trimethyl-14-pentadecanone, undecanal, nonanal, (E)-2-nonenal, (E)-2-decenal, dodecane, tridecane, and 2-pentylfuran (Figure 2). The deterioration degree of brown rice quality during storage can be reflected by the change in FAV, shown as an important index for measuring the freshness and aging degree of brown rice [6]. This suggested that the colorimetric values, 1-octanol, 1-nonanol, 2-hexyl-1-decanol, 2,6,10-trimethyl-14-pentadecanone, undecanal, nonanal, (E)-2-nonenal, (E)-2-decenal, dodecane, tridecane, and 2-pentylfuran, might indirectly reflect the storage quality of high-moisture japonica brown rice. It has been pointed out that the quality of cashew oil could be reflected by the color value [33]. Similarly, Liu et al. [34] also found that undecanal, nonanal, 2-decenal, and 2-pentylfuran were potential markers for evaluating lipid oxidation in fried soybean oil.

### 3.5. Analysis of Lipid Metabolism in High-Moisture Japonica Brown Rice at Different Storage Temperatures

In order to investigate the effect of temperature on the lipid metabolism in the high-moisture japonica brown rice, no-targeted lipidomics were used to identify the lipid metabolites in the high-moisture japonica brown rice. LC-MS identified 659 lipid metabolites under positive ionization mode and 235 lipid metabolites under negative ionization mode. Based on the whole lipid metabolism data, the effect of storage temperature on the lipid metabolism of the high-moisture japonica brown rice was revealed by the PCA and OPLS-DA methods. In the positive ionization mode (Figure 3A), the first and second principal components explained 17.5% (PC1) and 13.7% (PC2) of the total variance at 15 °C (red dot), 20 °C (blue dot), and 25 °C (green dot). In the negative ionization mode (Figure 3B), the first and second principal components explained 35.1% (PC1) and 18.6% (PC2) of the total variance at 15 °C (red dot), 20 °C (blue dot), and 25 °C (green dot). The results of PCA showed no significant difference between the high-moisture japonica brown rice stored at the three temperatures. In order to better distinguish statistically significant differences in the high-moisture japonica brown rice in different groups, the supervised statistical method OPLS-DA was introduced. Apparently, the high-moisture japonica brown rice stored at 15 °C, 20 °C, and 25 °C was well differentiated (Figure 3C–H) and all the samples were in the 95% confidence interval.

### 3.6. Identification of Key Lipid Metabolites in High-Moisture Japonica Brown Rice

To show the overall distribution of the lipid metabolites more clearly, the −log2 (FC) and −log10 *p* volcanoes for various substances are shown in Figure 4A–C. Metabolites satisfying VIP > 1 (OPLS-DA) and *p* < 0.05 (Student’s *t*-test) were screened based on the statistical results and labeled in red (up-regulated) and blue (down-regulated). In total, 25 metabolites were significantly down-regulated and 25 metabolites were significantly up-regulated at 15 °C vs. 20 °C. Similarly, 47 metabolites were significantly reduced while 30 metabolites were significantly increased at 15 °C vs. 25 °C. A total of 25 metabolites were significantly reduced while 13 metabolites were significantly increased at 20 °C vs. 25 °C. To better visualize the results, we used heatmaps to show the 30 metabolites with the largest differences in VIP values within each group (Figure 4D–F).

### 3.7. Analysis of Key Lipid Metabolism Pathways in High-Moisture Japonica Brown Rice

To elucidate the effect of temperature on the lipid metabolism of the high-moisture japonica brown rice, the significantly different metabolites (VIP > 1, *p* < 0.05) were imported into the KEGG database and matched with their KEGG ID information for a metabolic pathway analysis (Figure 5). The metabolic pathway analyses showed that these significantly different lipid metabolites were mainly involved in glycerophospholipid metabolism, glycerolipid metabolism, and sphingolipid metabolism.

#### 3.7.1. Glycerolipid Metabolism

Diglycerides (DG) are one of the major lipid subclasses in living organisms and second messengers in various cellular activities [35]. DG has an important role in glycerolipid metabolism and glycerophospholipid metabolism. DG was significantly down-regulated in the high-moisture japonica brown rice among rhw groups of 15 °C vs. 20 °C, 15 °C vs. 25 °C, and 20 °C vs. 25 °C under low-temperature (15 °C and 20 °C) storage (Figure 5). Correspondingly, the up-regulation of DG at 25 °C might have been caused by the breakdown of glycerophospholipids [36]. Notably, the change trends of OAcyl-(gamma-hydroxy) FA (OAHFA) were the same as DG. In previous studies, OAHFA was found to be a group of polar lipids in meibum, serving as a surfactant in the tear film lipid layer [37]. We speculated that OAHFA might bind to some proteins to participate in cell signal transduction, thereby affecting the storage process of brown rice under a low temperature.

Triglycerides (TG) are major lipids present in rice germ and rice bran [38]. Unlike rice, the bran of brown rice is partially destroyed during the hulling process, resulting in enzymes (lipase and lipoxygenase, etc.) coming into contact with TG, which undergo hydrolysis to form free fatty acids or are oxidized to form volatile oxidative degradation products through the enzymatic oxidation of lipoxygenase, generating a large number of flavor substances that can reduce the quality of brown rice [19]. However, TG was significantly up-regulated in the high-moisture japonica brown rice among the groups of 15 °C vs. 20 °C, 15 °C vs. 25 °C, and 20 °C vs. 25 °C under low-temperature (15 °C and 20 °C) storage (Figure 5). It could be concluded that low-temperature (15 °C and 20 °C) storage inhibited the hydrolysis of TG in the brown rice, thus decreasing the contents of free fatty acids and volatile oxidative degradation products and delaying the quality deterioration.

#### 3.7.2. Glycerophospholipid Metabolism

Glycerophospholipids (GL) are major structural components of cell membranes and play key roles in maintaining cellular homeostasis [39], mainly including phosphatidylethanolamine (PE) and phosphatidylserine (PS), which were identified in the study. PS is extremely heterogeneously distributed in the cell membrane, where it regulates cellular activity, can mediate apoptosis, and is important for cellular functions [40]. PE is the second-most abundant glycerophospholipid in eukaryotic cells and has significant biological activities such as antioxidant, antimicrobial, and the regulation of lipid metabolism [41]. PS and PE were significantly up-regulated in the high-moisture japonica brown rice stored at 15 °C in both groups, 15 °C vs. 20 °C and 15 °C vs. 25 °C (Figure 5). Similarly, after storing rice grains at 20 °C for 360 and 540 days, Zhang et al. [35] found that the PE and PS contents of the rice grains decreased during storage. In fact, the products of hydrolysis of glycerophospholipids will lead to the deterioration of the edible and nutritional quality of brown rice [42]. This suggested that storage at 15 °C could delay the hydrolysis of glycerophospholipids in high-moisture japonica brown rice, thereby delaying the deterioration of its storage quality.

#### 3.7.3. Sphingolipid Metabolism

Ceramides (Cer) and glycosylceramides (Hex1Cer) are components of sphingolipids and Glycosphingolipid, which are important lipids in many cellular processes in life; these molecules are key structural compounds in the lipid bilayer of cell membranes [43]. Cer and Hex1Cer were significantly down-regulated in the high-moisture japonica brown rice stored at low temperatures (15 °C and 20 °C) among the groups of 15 °C vs. 20 °C, 15 °C vs. 25 °C, and 20 °C vs. 25 °C (Figure 5). In contrast, high-moisture japonica brown rice showed a significant up-regulation of Cer and Hex1Cer at the end of 90 days of storage at 25 °C. Similarly, Liu et al. [13] also found a significant increase in cer and hex1cer content after storing eggs at 22 °C for 28 days. The increased cer and hex1cer content might have been induced by lipid decomposition, main chain cleavage, sidechain modifications, or lipolysis during storage. In addition, ethanolamine phosphate (PEA) could be involved in glycerophospholipid metabolism after production by Cer (Figure 5). It was inferred that low-temperature (15 °C and 20 °C) storage could reduce the breakdown of glycerophospholipids by slowing down the sphingolipid metabolism, which could effectively maintain the edible quality and nutritional quality of high-moisture japonica brown rice.

## 4. Conclusions

In this study, the effect of temperature on the storage quality of high-moisture japonica brown rice was investigated. In addition, the lipid metabolites of high-moisture japonica brown rice were researched using an untargeted lipidomics approach, which could explain the mechanism of low-temperature storage to delay the quality deterioration of brown rice. After storing high-moisture japonica brown rice at 15 °C, 20 °C, and 25 °C for 90 days, we found that the high-moisture japonica brown rice stored at 15 °C maintained a preferable color, lower content of fatty acid values, and better retention of volatile compounds than at the other two temperatures. In addition, storage at 15 °C could delay the decomposition of glycerophospholipids as well as the metabolism of glycerolipid and sphingolipids in brown rice during storage. Therefore, the low-temperature environment regulated the glycerolipid, sphingolipid, and glycerophospholipid catabolism pathways to maintain higher contents of TG, PS, and PE, and lower contents of DG, OAHFA, Cer, and Hex1Cer in the high-moisture japonica brown rice. This study provides a better understanding for the different changes among three storage temperatures for tje lipid metabolism in brown rice.

## Figures and Tables

**Figure 1 foods-12-04218-f001:**
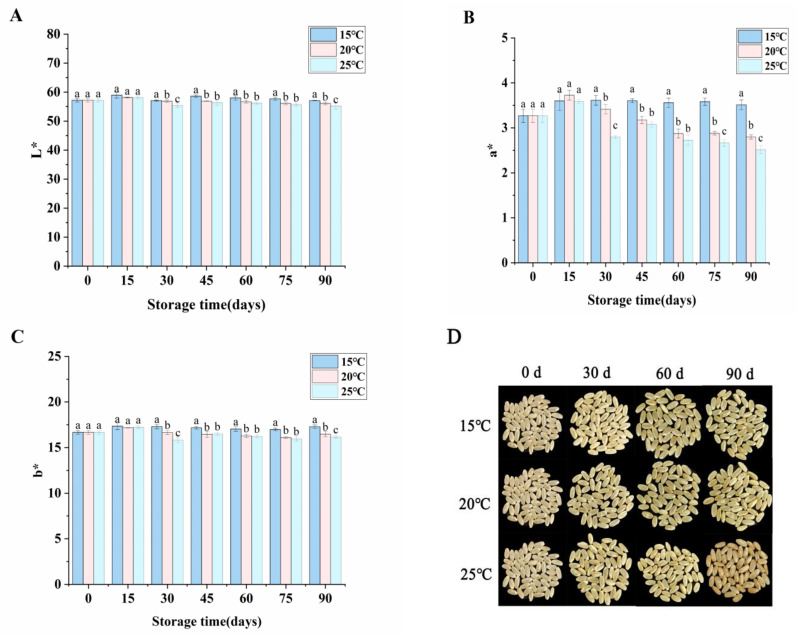
Changes in L* (**A**), a* (**B**), b* (**C**), appearance (**D**), FAV (**E**), and volatile compound types (**F**) of high-moisture japonica brown rice during storage. Different lowercase letters indicated significant differences (*p* < 0.05).

**Figure 2 foods-12-04218-f002:**
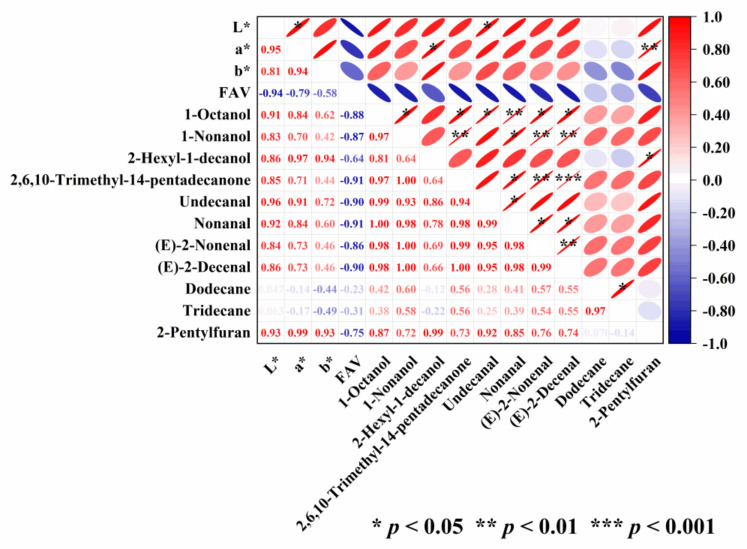
Heat map of correlation among color, FAV, and volatile compounds.

**Figure 3 foods-12-04218-f003:**
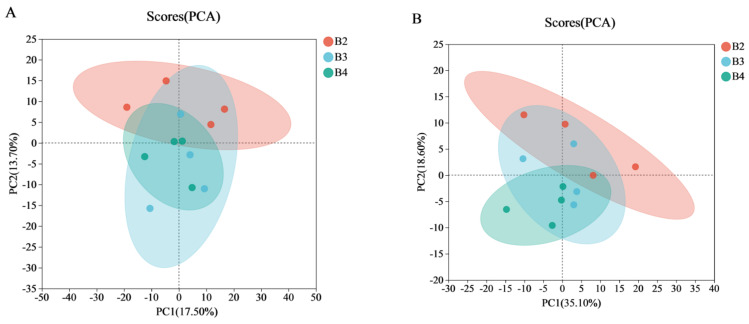
(**A**,**B**) PCA score plots of the different groups in positive and negative ionization mode. (**C**,**D**) OPLS-DA score plots of 15 °C vs. 20 °C in positive and negative ionization mode. (**E**,**F**) OPLS-DA score plots of 15 °C vs. 25 °C in positive and negative ionization mode. (**G**,**H**) OPLS-DA score plots of 20 °C vs. 25 °C in positive and negative ionization mode. The storage temperatures of high-moisture japonica brown rice were denoted by the letters B2, B3, and B4, representing 15 °C, 20 °C, and 25 °C, respectively.

**Figure 4 foods-12-04218-f004:**
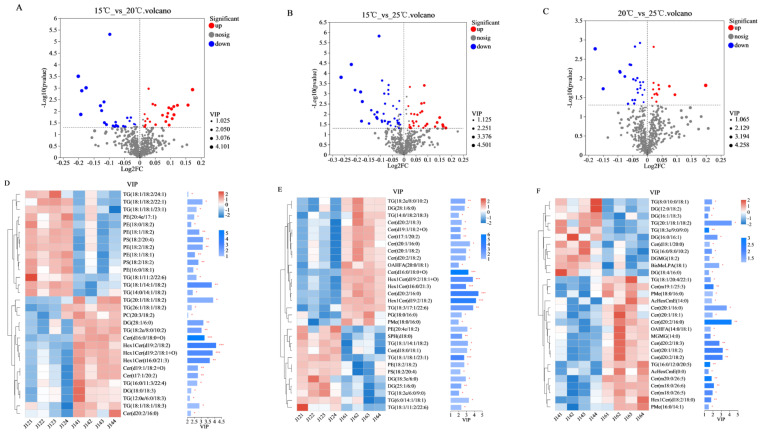
Volcano plots and heatmaps in high-moisture japonica brown rice stored at 15 °C, 20 °C, and 25 °C at 90 days. Volcano plots of 15 °C vs. 20 °C (**A**), 15 °C vs. 25 °C (**B**), and 20 °C vs. 25 °C (**C**). Heatmaps of VIP values of significantly different metabolites in 15 °C vs. 20 °C (**D**), 15 °C vs. 25 °C (**E**), and 20 °C vs. 25 °C (**F**). In the volcano plot, significantly up-regulated metabolites, down-regulated metabolites, and non-significantly regulated metabolites are shown in red, blue, and gray, respectively. In the clustering heat map, red and blue colors indicate higher or lower metabolite contents, respectively; high-moisture japonica brown rice stored at 15 °C is J121, J122, J123, and J124; high-moisture japonica brown rice stored at 20 °C is J141, J142, J143, and J144; and high-moisture japonica brown rice stored at 25 °C is J161, J162, J163, and J164. In addition, * means *p* < 0.05, ** means *p* < 0.01, *** means *p* < 0.001.

**Figure 5 foods-12-04218-f005:**
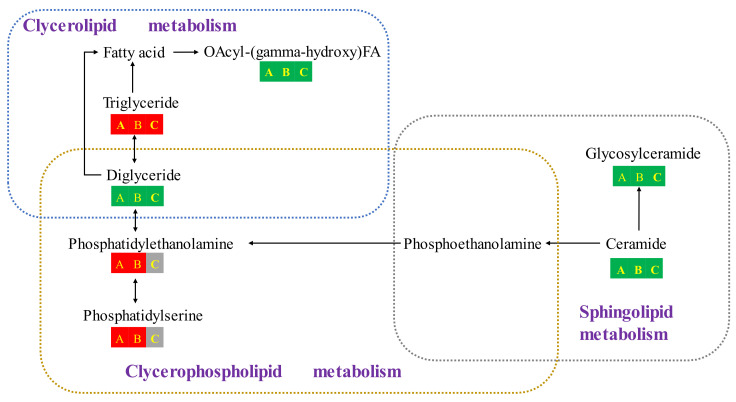
Changes in metabolites in the metabolic pathways of high-moisture japonica brown rice at the three storage temperatures. A, B, and C in the figure represent the treatment groups of 15 °C vs. 20 °C, 15 °C vs. 25 °C, and 20 °C vs. 25 °C, respectively. The red blocks represent the up-regulated metabolites of high-moisture japonica brown rice at a low temperature compared to a high temperature. The green block represents the down-regulated metabolites of high-moisture japonica brown rice at a low temperature compared to a high temperature. The gray blocks represent that the lipid metabolites of brown rice were not significant at low temperatures compared to high temperatures.

**Table 1 foods-12-04218-t001:** Concentration of volatile compounds (%) in high-moisture japonica brown rice with HS-SPME-GC/MS analysis.

NO.	Volatile Compounds	15.5%			
		0 Day	90 Day		
		-	15 °C	20 °C	25 °C
Alcohols					
1	1-Heptanol	1.89 ± 0.13 ^a^	1.86 ± 0.12 ^a^	/	0.33 ± 0.02 ^b^
2	1-Octanol	5.25 ± 0.20 ^a^	3.99 ± 0.19 ^b^	1.93 ± 0.12 ^c^	1.70 ± 0.44 ^c^
3	1-Nonanol	1.46 ± 0.12 ^a^	0.95 ± 0.05 ^b^	0.64 ± 0.08 ^c^	0.60 ± 0.08 ^c^
4	1-Octen-3-ol	/	5.72 ± 0.03 ^a^	5.43 ± 0.04 ^b^	5.15 ± 0.03 ^c^
5	2-Octyl-1-decanol	0.44 ± 0.07 ^a^	0.47 ± 0.08 ^a^	/	/
6	2-Hexyl-1-decanol	1.87 ± 0.17 ^a^	2.01 ± 0.41 ^a^	1.12 ± 0.22 ^b^	1.10 ± 0.33 ^b^
7	Linalool oxide	/	0.96 ± 0.02 ^a^	0.84 ± 0.22 ^a^	0.39 ± 0.07 ^b^
8	alpha-Terpineol	/	1.76 ± 0.04 ^a^	1.19 ± 0.12 ^b^	0.58 ± 0.25 ^c^
9	4,8-Dimethyl-1-nonanol	0.83 ± 0.06 ^a^	0.76 ± 0.01 ^a^	0.45 ± 0.09 ^b^	0.25 ± 0.06 ^c^
10	3,7,11-Trimethyl-1-dodecanol	0.23 ± 0.02 ^b^	0.38 ± 0.03 ^a^	/	0.26 ± 0.04 ^b^
Ketones					
11	Geranylacetone	1.75 ± 0.15 ^a^	1.19 ± 0.10 ^b^	/	/
12	(R,S)-5-Ethyl-6-methyl-3E-hepten-2-one	0.97 ± 0.23 ^a^	0.86 ± 0.05 ^a^	0.41 ± 0.18 ^b^	0.20 ± 0.04 ^b^
13	6,10-Dimethyl-undecan-2-one	0.98 ± 0.06 ^a^	1.02 ± 0.13 ^a^	0.51 ± 0.11 ^b^	/
14	2,6,10-Trimethyl-14-pentadecanone	2.36 ± 0.45 ^a^	1.20 ± 0.08 ^b^	0.59 ± 0.05 ^c^	0.31 ± 0.10 ^c^
Aldehydes					
15	Undecanal	0.68 ± 0.01 ^a^	0.59 ± 0.01 ^a^	0.35 ± 0.06 ^b^	0.28 ± 0.09 ^b^
16	Tetradecanal	0.44 ± 0.07 ^a^	0.32 ± 0.03 ^b^	/	/
17	Octanal	2.17 ± 0.31 ^a^	2.29 ± 0.25 ^a^	/	/
18	Nonanal	14.60 ± 0.21 ^a^	11.94 ± 0.68 ^b^	8.56 ± 0.41 ^c^	7.59 ± 1.47 ^c^
19	Decanal	3.32 ± 0.09 ^a^	3.05 ± 0.05 ^b^	/	/
20	(E)-2-octenal	2.17 ± 0.10 ^a^	1.00 ± 0.35 ^b^	/	/
21	(E)-2-Nonenal	1.29 ± 0.08 ^a^	0.70 ± 0.15 ^b^	0.22 ± 0.00 ^c^	0.21 ± 0.03 ^c^
22	(E)-2-Decenal	4.71 ± 0.54 ^a^	2.48 ± 0.33 ^b^	1.10 ± 0.11 ^c^	0.64 ± 0.03 ^d^
23	2-Undecenal	3.88 ± 0.46 ^a^	2.32 ± 0.18 ^b^	/	/
Hydrocarbons					
24	Dodecane	1.88 ± 0.06 ^a^	1.11 ± 0.03 ^b^	1.20 ± 0.03 ^b^	1.57 ± 0.04 ^a^
25	Tridecane	2.53 ± 0.42 ^a^	0.98 ± 0.02 ^c^	1.48 ± 0.15 ^b^	1.81 ± 0.17 ^b^
26	Tetradecane	1.93 ± 0.35 ^c^	2.28 ± 0.05 ^b^	2.54 ± 0.02 ^a^	2.78 ± 0.21 ^a^
27	Heptadecane	0.20 ± 0.01 ^d^	0.69 ± 0.00 ^c^	1.09 ± 0.02 ^b^	1.50 ± 0.04 ^a^
28	Tetradecane, 3-methyl-	/	0.50 ± 0.03 ^b^	1.19 ± 0.62 ^a^	1.53 ± 0.35 ^a^
29	Pentadecane, 3-methyl-	/	1.01 ± 0.02 ^b^	1.21 ± 0.05 ^a^	1.27 ± 0.01 ^a^
30	Pentadecane, 4-methyl-	/	/	0.43 ± 0.18 ^b^	1.16 ± 0.24 ^a^
31	Pentadecane, 5-methyl-	/	0.61 ± 0.01 ^b^	0.95 ± 0.07 ^a^	1.18 ± 0.21 ^a^
32	Cyclopentane, undecyl-	0.28 ± 0.10 ^c^	0.66 ± 0.01 ^b^	0.81 ± 0.02 ^a^	0.86 ± 0.05 ^a^
33	Dodecane, 4,6-dimethyl-	/	0.54 ± 0.02 ^b^	0.57 ± 0.03 ^b^	0.68 ± 0.01 ^a^
34	Hexadecane, 2,6,10,14-tetramethyl-	/	0.64 ± 0.05 ^b^	1.08 ± 0.41 ^a^	1.28 ± 0.14 ^a^
35	Heneicosane, 11-(1-ethylpropyl)-	/	/	0.89 ± 0.67 ^a^	1.32 ± 0.03 ^a^
Others					
36	2-pentylfuran	4.90 ± 0.04 ^b^	5.98 ± 0.32 ^a^	2.25 ± 0.45 ^c^	1.53 ± 0.57 ^c^
37	Ethyl palmitate	0.38 ± 0.00 ^b^	0.60 ± 0.06 ^a^	/	/
38	2,4-bis(1,1-dimethylethyl)- phenol	/	2.18 ± 0.09 ^a^	1.38 ± 0.10 ^b^	0.92 ± 0.04 ^c^

Note: “/” stands for not detected. Values in the same row with different superscript letters were significantly different (*p* < 0.05).

## Data Availability

The article contains all the relevant data. The original contributions presented in the study are included in the article; further inquiries can be directed to the corresponding author.

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
