# Peer review of "Untargeted Lipidomics Reveal Quality Changes in High-Moisture Japonica Brown Rice at Different Storage Temperatures"

_foods, 2023, doi:10.3390/foods12234218_

Round 1

Reviewer 1 Report

Comments and Suggestions for Authors

In the paper was given the effect of temperature on the storage quality of high-moisture japonica brown rice. The authors made a concise experimental design and appropriate statistical analysis, according myself. The results are decently presented, althrough I think it could be done better. Hence, I would like to suggest expanding the section where statistical analysis was given. I find these descriptions insufficient. I must make an acknowledgement to authors, the oldest reference is six years old. Congratulations for searching the most recent literature what is a huge work by itself. Therefore, I would like to suggest a minor revision mainly related to interpretation of statistical analysis sections. 

Reviewer 2 Report

Comments and Suggestions for Authors

The manuscript is well-written and scientifically sound. However, I need to confirm several issues.

1. In the introduction. How the authors set the low storage temperatures at 15, 20, and 25C is not clear. why 90 days? This description should be moved to materials and methods.

2. Color measurement for single-kernel rice? or bulk? not clear. Did the authors measure the white reference for color measurement? Every how many days of color measurement? every 5 days? Not clear.

3. Please show in one Figure the real condition of storage experimentation. How about the RH during the storage?

4. For nondestructive parameters such as color information, the measurement can be made on the same samples. However, for destructive parameters such as fatty acids or lipidomic, the measurement can not be made on the same sample during 90 days of storage. It is not clear how the authors did the sampling. Only on 0 and 90 days? 

Round 2

Reviewer 2 Report

Comments and Suggestions for Authors

The revision is complete. I accept this final version to be published in Foods.